# Gene Expression Patterns Associated with Survival in Glioblastoma

**DOI:** 10.3390/ijms25073668

**Published:** 2024-03-25

**Authors:** Christopher Morrison, Eric Weterings, Nicholas Gravbrot, Michael Hammer, Martin Weinand, Abhay Sanan, Ritu Pandey, Daruka Mahadevan, Baldassarre Stea

**Affiliations:** 1Department of Radiation Oncology, University of Arizona, Tucson, AZ 85719, USA; chris.morrison@arizona.edu (C.M.);; 2College of Medicine, University of Arizona, Tucson Campus, Tucson, AZ 85724, USA; nicholas.gravbrot@hci.utah.edu (N.G.); mfh@arizona.edu (M.H.); 3Department of Neurology, University of Arizona Genetics Core, Tucson, AZ 85724, USA; 4Department of Neurosurgery, University of Arizona, Tucson, AZ 85724, USA; mweinand@arizona.edu; 5Center for Neurosciences, Tucson, AZ 85719, USA; asanan@neurotucson.com; 6Department of Cellular and Molecular Medicine, University of Arizona Cancer Center Bioinformatics Shared Resource, and College of Medicine, University of Arizona, Tucson, AZ 85724, USA; ritu@arizona.edu; 7Mays Cancer Center, University of Texas Health, San Antonio, TX 78229, USA; mahadevand@uthscsa.edu

**Keywords:** glioblastoma, differential gene expression, MGMT, glioblastoma prognostic index, TCGA

## Abstract

The aim of this study was to investigate gene expression alterations associated with overall survival (OS) in glioblastoma (GBM). Using the Nanostring nCounter platform, we identified four genes (*COL1A2, IGFBP3, NGFR*, and *WIF1*) that achieved statistical significance when comparing GBM with non-neoplastic brain tissue. The four genes were included in a multivariate Cox Proportional Hazard model, along with age, extent of resection, and O6-methylguanine-DNA methyltransferase (*MGMT)* promotor methylation, to create a unique glioblastoma prognostic index (GPI). The GPI score inversely correlated with survival: patient with a high GPI had a median OS of 7.5 months (18-month OS = 9.7%) whereas patients with a low GPI had a median OS of 20.1 months (18-month OS = 54.5%; log rank *p*-value = 0.004). The GPI score was then validated in 188 GBM patients from The Cancer Genome Atlas (TCGA) from a national data base; similarly, patients with a high GPI had a median OS of 10.5 months (18-month OS = 12.4%) versus 16.9 months (18-month OS = 41.5%) for low GPI (log rank *p*-value = 0.0003). We conclude that this novel mRNA-based prognostic index could be useful in classifying GBM patients into risk groups and refine prognosis estimates to better inform treatment decisions or stratification into clinical trials.

## 1. Introduction

Glioblastoma (GBM), a WHO grade IV astrocytoma, is the most commonly diagnosed primary brain tumor in adults. The disease is characterized by inherent invasiveness beyond the surgical bed, rapid progression, and high treatment resistance [1]. Little progress has been made in the treatment of GBM since the NCIC/EORTC trial by Stupp et al. defined the current standard of care, which consists of maximal safe resection followed by adjuvant temozolomide (TMZ) and 60 Gy of fractionated radiation [2]. At present, this standard of care is not curative for most patients and median overall survival (OS) is approximately 14 months [2]. The recent addition of electrical fields (Optune, Novocure, Root, Switzerland) to the Stupp regimen has shown a promising improvement in median OS by 5 months [3], but this treatment modality has several barriers to wide-spread implementation [4].

In addition to established clinical prognostic factors, such as age, tumor size and location, extent of resection, and Karnowsky Performance Score (KPS) [5,6,7], a few genetic markers are currently used to inform prognosis. Most notably, mutations in the isocitrate dehydrogenase (IDH) gene predict for a longer OS, while telomerase reverse transcriptase (TERT) promoter mutations predict a shorter OS [8]. In addition, epigenetic silencing of O^6^-methylguanine-DNA methyltransferase (MGMT) is both prognostic and predictive of an increased response to TMZ and thus a longer OS [9]. 

Despite the information obtained from clinical features and genetic/epigenetic changes, it is difficult to estimate survival after treatment for individual patients; in addition, GBM patients can have widely different clinical courses after nearly identical treatments. Therefore, we undertook this retrospective study of genomic analysis of GBM specimens (obtained from patients at the time of initial tumor resection), correlated their mRNA expression profiles to patients’ survival and finally derived an individual prognostic index that combines all know prognostic indicators with the differential expression of four new genes. Because the nature of this study was retrospective, we needed to use a gene expression assay capable of accurately quantifying mRNA levels in formalin-fixed, paraffin embedded (FFPE) tissue samples, as this is the most widely used long-term storage method for pathological samples. To meet this goal, the NanoString nCounter (Nanostring Technologies, Seattle, WA, USA) platform was chosen because the nCounter platform has already been proven to be as reliable as other methods of gene expression quantification—microarrays, RT-PCR, and RNA-Seq—with Formalin Fixed Paraffin Embedded (FFPE) tissue samples [10,11,12]. The goal of this study was to search for biologic differences that might correlate with overall survival after standard treatment with surgery, chemo-radiation (chemo-RT), and adjuvant TMZ chemotherapy. Using comparative differential expression analyses combined with multivariate Cox proportional hazards (CPH) models, we have derived a prognostic index that can discriminate between GBM patients with significantly different OS expectations. We then validated the utility of this prognostic index within an independent, publicly available dataset from The Cancer Genome Atlas (TCGA) [13]. Our research hypothesis was that there may be a set of genes involved in generating resistance to chemotherapy and radiation therapy and that along with traditional prognostic factors (age, KPS, extent of resection, etc.) they might be better able to predict survival after standard therapy (Stupp protocol) and therefore better inform patients and physicians alike of the prognosis of individual patients with GBM.

## 2. Results

### 2.1. GBM Patients, Tumor, and Treatment Characteristics

The study population consisted of 23 patients (17 males and 6 females) with a median age of 64 years. The median OS of the 23 GBM patients in our institutional study group was 11.4 months. All 23 patients were uniformly treated with a maximal safe resection followed by chemo-RT with 60 Gy and at least 6 months of adjuvant TMZ. Patients who underwent a biopsy or did not complete the full course of conventionally fractionated radiation were not included in this study. We stratified the 23 GBM patients in two subgroups based on their median survival: a ‘Short OS’ group, which included patients with a survival less than or equal to the cohort’s median OS, and a ‘Long OS’ group, with an OS greater than the cohort’s median survival. A comparison of the demographic, tumor, and treatment characteristics between the ‘Long OS’ and ‘Short OS’ groups is shown in Table 1A. No significant differences in baseline characteristics were found between the two groups, and even the gender proportions between the two groups were not significantly different. Patients in the ‘Long OS’ group trended to be younger (median age = 63 vs. 70.5), and a higher proportion had undergone a GTR (45.5% vs. 16.7%), but these differences were not statistically significant (*p* = 0.19 for both comparisons). There were three patients in the ‘Long OS’ group and two in the ‘Short OS’ group with MGMT promotor methylation. The differential expression of the MGMT gene was also not statistically different between the two groups (*p* = 0.08).

### 2.2. Differential Expression of Pathways and Individual Genes

To arrive at a set of differentially expressed genes, we first compared gene expression profiles between the 23 GBM patients and the 12 temporal lobe tissue samples obtained from epilepsy surgery, which were used as non-neoplastic control. Differential expression (DE) analysis of individual genes identified 315 genes with >2-fold DE and 25 genes with >16-fold DE in the GBM vs. control tissue comparison (all adjusted *p*-values < 0.01). Of those 25 genes with >16-fold DE, notably WIF1 was the only downregulated gene, whereas the other 24 genes displayed increased expression in GBM tissue relative to control.

Next, we compared the DE profiles between GBM patients in the ‘Long OS’ group and the ‘Short OS’ group and we found that 17 genes displayed > 2-fold DE in that comparison (adjusted *p*-values < 0.05). Cross-referencing the GBM vs. non-neoplastic brain with the ‘Long OS’ vs. ‘Short OS’ DE profiles, identified 12 of the 17 genes that were common to both comparisons. Table 2 lists these 12 dysregulated genes, the difference in expression levels between groups and the biochemical pathway affected.

We also found significant dysregulation of the 13 pathways defined by the nCounter PanCancer Human Pathways Panel. A global overview is presented in the form of heatmaps of pathway scores shown in Figure 1. The first panel (Figure 1A) shows the comparison of pathway scores between the non-neoplastic brain tissue and all 23 GBM tissues, while the second (Figure 1C) shows the Long versus Short OS comparison for only the GBM samples in the study group. These pathway scores are then summarized as a single pathway signature for each comparison group (Figure 1B,D). In addition to the pathway signatures, the directed global significance scores also provide a global summary measure of the overall differential expression of each pathway, and those are shown in Table 3. The heatmap of GBM versus control brain tissue (Figure 1A) clearly shows a homogenous expression profile amongst the control brain tissues with consistent patterns of upregulation in about half of the pathways and downregulation in the other half of the pathways when the GBM tissues are compared to controls. Most notably, the Wnt pathway was found to be the most negatively dysregulated pathway in the comparison of GBM tissue versus the non-neoplastic controls, while the Notch pathway was strongly positively dysregulated. However, when comparing the Short vs. Long OS GBM groups, only 2 of the 13 pathways were upregulated (DNA repair and Transcriptional Dysregulation) while the other 11 pathways were negatively affected (Table 3).

### 2.3. Cox Proportional Hazards Analysis and Derivation of Glioblastoma Prognostic Index

Among the 12 genes identified in the combined DE analysis, we focused on the 6 genes that had both more than an 8-fold DE in the GBM vs. controls and more than a 3-fold DE difference in the ‘Short OS’ vs. ‘Long OS’ comparison. These six genes were MMP9, IBSP, COL1A2, IGFBP3, NGFR, and WIF1. We then performed univariate CPH models for the well-known clinical prognostic factors of age, KPS, tumor size, extent of surgical resection, and MGMT promotor methylation status, as well as these six highly differentially expressed genes (Table 4). Next, we performed a multivariate CPH analysis with the clinical and genomic factors with a *p*-value < 0.1 on the univariate analysis, resulting in the following prognostic factors: age, extent of resection, and four of the six genes. We also included MGMT promoter methylation status in the multivariate model, despite it not meeting this *p*-value inclusion criteria, because of its well-established and significant prognostic implications in larger datasets. The regression coefficients obtained from this multivariate CPH model (shown in the right-most column of Table 4) were then used to create a weighted sum of each factor included in the multivariate CPH to create a ‘Glioblastoma Prognostic Index’ (GPI). The GPI is then calculated as follows: GPI = 0.074 × Age − 0.101 × WIF1 + 0.448 × NGFR + 0.09 × IGFBP3 + 0.242 × COL1A2 + 1.43 if an STR and −0.876 if MGMT methylated.

### 2.4. GPI Survival Analysis for the Study Group

The entire cohort of 23 GBM patients was then stratified into two groups of patients, one group with their GPI below the entire cohort’s median GPI (‘Low GPI’ group) and one group with their GPI greater than or equal to the median (‘High GPI’ group). Kaplan–Meier (KM) analysis was then used to compare the OS probability between the ‘Low GPI’ and ‘High GPI’ groups. While the median OS for the entire cohort was 11.4 months, the ‘High GPI’ group was found to have a median OS of 7.5 months, significantly shorter than the ‘Low GPI’ group, which had a median OS almost 3 times greater at 20.1 months (log-rank *p* value = 0.004) (Figure 2A). Of note, we found this methodology to be significantly better than using MGMT methylation status alone, which resulted in a non-significant difference in OS estimates of 11.4 months for non-methylated patients versus 15.7 months for methylated patients (log-rank *p* = 0.3).

### 2.5. Validation of the GPI in the TCGA GBM Cohort

A comparison of the clinical characteristics between the original 23 GBM patients in the study group and the TCGA cohort used for external validation of the GPI is shown above in Table 1B. There were no statistically significant differences between the two groups except that the proportion of patients with MGMT promotor methylation was significantly higher in the TCGA cohort (44.7% vs. 21.7%; *p* = 0.04). The median OS of the study group cohort was 11.4 months while the median OS for the TCGA cohort was 8.0 months, a difference that was not statistically significant, likely a result of the fact that 25.5% of the TCGA patients had unknown KPS values and there were no data available to us regarding the extent of resection or details regarding their adjuvant therapies (dose of radiation and administration of chemotherapy).

As was carried out for our study group, we classified TCGA GBM patients into two risk groups based on their Z-score GPI and compared their KM OS estimates. Figure 2B shows that within the TCGA cohort of patients, the GPI was also able to distinguish between patients with statistically significant differences in OS, despite not having data on the extent of resection, a heavily weighted factor in the GPI. TCGA patients with a high GPI had a median OS of 10.5 months with an estimated 18-month OS of 12.4%, while patients with a low GPI had median OS of 16.9 months and a 41.5% 18-month OS (log rank *p* value = 0.0003). As was seen in our study group, the separation of survival expectations within the TCGA is greater when using the GPI as compared to using MGMT methylation status alone (median OS = 11.7 months when non-methylated vs. 14.0 months for methylated; log-rank *p* = 0.3). This is further illustrated in Figure 2C by plotting the Kaplan–Meier OS estimates for the TCGA cohort grouped by both MGMT methylation status and GPI. The additional value the GPI provides over MGMT status alone comes mostly from identifying a sub-group of MGMT methylated patients with very poor prognosis (median OS = 8.8 months for MGMT methylated patients with a high GPI). Patients with low GPI scores have a relatively good prognosis regardless of MGMT status (median OS = 16.9 months for unmethylated vs. 17.8 months for methylated).

## 3. Discussion

We have derived a novel composite biomarker for patients with GBM, the GBM prognostic index (GPI), based on known prognostic clinical variables (patient age, resection status), and the expression levels of four genes; the GPI score is inversely proportional to the survival and can significantly refine OS prediction in GBM patients undergoing treatment with the current standard of care (maximal safe resection followed by the Stupp regimen). This index’s prognostic power is shown by the median OS of patients with a low GPI being nearly three times that of patients with a high GPI (20.1 months vs. 7.5 months (Figure 2A). The findings derived from our small cohort of 23 homogeneously treated patients were then validated with an independent gene expression dataset from a larger group of patients from TCGA showing the same clinically meaningful differences in the OS of patients based on their GPI (41.5% vs. 12.4% 18-month OS for low vs. high GPI respectively), although in the larger dataset, the difference between the ‘long’ and the ‘short’ groups was smaller, likely due to absence of details for the extent of resection and use/dose of chemoRT (Figure 2B). The usefulness of the novel GPI resides in its potential to refine our prognostication abilities, which could improve the patient and physician decision-making process regarding optimal adjuvant treatment after maximal safe resection as well participation in clinical trials. For instance, patients with a high GPI (worse prognosis) could elect for a less burdensome, hypo-fractionated radiation treatment course, such as those that have already been shown to be non-inferior to the standard Stupp-based regimen in subsets of patients with poor prognosis [14,15], or patients with a low GPI (better prognosis) could elect to enroll in more aggressive investigational treatment protocols following the standard regimen. One strength of this study was the use of a gene expression assay that can be run on FFPE tissue samples, rather than fresh or frozen tissue samples, as FFPE tissues are much more readily available for retrospective analyses and much easier to work with in standard day-to-day clinical operations. There are numerous published analyses of GBM genomics resulting in classification systems [16] or comparisons between long and short survivors [17,18,19], but to our knowledge none have directly predicted prognosis using a combination of well-established clinical factors and genomic data acquired from tissue samples. Yin et al. created a gene expression-based prognostic score for GBM [20] that is similar to the GPI, but their algorithm neglects important clinical factors, like age and extent of resection, and is derived from the TCGA microarray data and thus would require fresh frozen tissue samples to be used clinically, something that is logistically difficult to do in actual daily practice.

The weakness of this investigation is in some key differences between our original cohort of 23 patients and the TCGA cohort of 188 patients used for validation. The first is the technical difference in how gene expression was quantified for each cohort. Our samples were analyzed using NanoString Technologies’ nCounter platform (Seattle, WA, USA) while the gene expression levels in TCGA cohort used in this study were derived from an Agilent microarray. These two methods resulted in different distributions of counts between the two assays, so a z-score transformation was necessary to account for these differences. While that transformation helps improve the applicability of our NanoString derived GPI to the TCGA dataset, there may still be important fundamental differences in the underlying mechanism of how each of these systems quantify gene expression. In addition to the technical differences, there are also important clinical differences between the two groups reported in this study. The biggest difference is the lack of data in the TCGA dataset on the extent of resection or dose of radiation therapy. This was a key component in the GPI derived from our original cohort, and without that information, the ability of the GPI to accurately estimate prognosis within the TCGA dataset could be hampered or diluted. Despite this missing piece of information, the GPI was still able to discriminate between patients with poor prognosis, regardless of their MGMT status. Another potential difference between our dataset and the confirmatory TCGA dataset is that our cohort of GBM patients comes from a single institution and thus our cohort of patients may have different underlying tumor biology than the presumably more heterogeneous TCGA cohort, due to the underlying ethnic composition of patients treated at our institution, or due to epigenetic changes induced by environmental factors unique to our area. Because of the difference in technology used to measure mRNA expression profiles and the small, potentially homogenous population of patients used to derive the GPI, it would be valuable to prospectively validate this new prognostic index in a larger, more heterogenous population of GBM patients, using the same nCounter quantification method as the original cohort of 23 patients.

In addition to helping improve prognostication at the time of diagnosis, the genes identified during this process may play important roles in the pathogenesis and/or treatment resistance of GBM. NGFR, or nerve growth factor receptor, is also known as the p75 neurotrophin receptor. As its name implies, NGFR is an important signaling receptor for the development, maintenance, and growth of both normal neural tissues and gliomas [21]. Nerve growth factors (NGFs), in concert with the NGFR receptor, have been found to stimulate the proliferation of glioma cell lines [21,22,23] at least in part through the deactivation of p53 [24]. It has also been implicated by multiple investigators to play a role in the invasiveness of gliomas [25,26]; this provides a strong rationale for the negative correlation between expression of this gene and survival.

IGFBP3 is a member of the insulin-like growth factor binding protein (IGFBP) family and has also been shown to play a role in the pathogenesis of diverse malignancies, including gliomas [27]. The IGFBP3 protein facilitates stabilization of insulin-like growth factors in circulation. The effects of IGFBP family dysregulation appear to be tumor dependent and studies have reported both tumor growth promoting and suppressing activity for IGFBP proteins [27]. In our study, high expression levels of IGFBP3 correlated with poor survival expectancy, and it corroborates the findings of two other studies [28,29]. Furthermore, it has also been demonstrated in vitro that suppression of IGFBP3 expression decreases proliferation rates of GBM cell culture and xenograft models [28].

WIF1 was the only gene in the GPI that was found to be downregulated in GBM tissues relative to the controls and also downregulated in patients with short OS relative to those with long OS. WIF1 is known to be a secreted antagonist of the Wnt pathway and, given that the Wnt pathway was also one of the most dysregulated pathways in our dataset (Table 2), the importance of WIF1 is even more plausible. Wnt signaling is already known to play an important role in glioma pathogenesis [30,31,32,33] by promoting glioma stem cell proliferation [33,34,35,36], driving glial-mesenchymal transitioning [37], and mediating resistance to both TMZ and radiation [33,38,39]. Furthermore, WIF1 expression has been found to be downregulated in astrocytomas, with the degree of WIF1 suppression correlating with the histological grade of the tumors [40,41,42]. In our study, WIF1 levels were down-regulated almost 20-fold in GBM tissue relative to non-neoplastic brain tissue, and over 3-fold down-regulated in short surviving GBM patients, relative to longer surviving counterparts (Table 3). Further investigation into this unique molecule seems warranted.

Finally, a recent publication has shed some light on the significance of the COL1A2 gene, the fourth gene in the GPI, showing that mRNA expression of COL1A2 (type 1 collagen) was significantly associated with worse prognosis in patients with diffuse glioma in the TCGA and CGGA cohorts (log rank *p* < 0.001) and although the study was specifically for diffuse low-grade gliomas, the genes involved were selected by an overlap with high-grade gliomas [43]. Thus, this new finding supports the inclusion of this new gene (COL1A2) in our novel GPI for GBM. Figure 3 summarizes the pathways affected by these four genes and the interactions amongst them.

## 4. Materials and Methods

### 4.1. Clinical Study Design, Patient Selection, and Clinical Data Collection

This study included 23 patients (study group) with a diagnosis of IDH1 wild-type GBM, who underwent maximal safe resection and completed conventionally fractionated adjuvant radiation (60 Gy) with concurrent TMZ at our institution. A retrospective review of these patients’ medical records was conducted to collect de-identified information on their demographics, treatment and tumor characteristics, and treatment outcomes. Overall survival was defined as the elapsed time between the date of the surgery that established the diagnosis and the date of death (from GBM or other cause). For patients who were still alive at the time of clinical data collection, the date of their last physician encounter or their last imaging procedure, whichever came later, was used to calculate OS. Tumor size (in square centimeters) was defined by multiplying the maximal axial anterior-posterior dimension by the maximal orthogonal lateral dimension on pre-operative contrast-enhanced T1 MRI. The extent of tumor resection was defined as either gross total resection (GTR) or subtotal resection (STR), based on the assessment of post-operative MRI by board-certified neuroradiologists. KPS groups were defined as either ≥70 or <70. As a control group in identifying informative differentially expressed genes, we utilized archived non-neoplastic cerebral cortical tissue obtained from 12 patients who underwent temporal lobe resection as part of their treatment for epilepsy. Lateral temporal cortical tissue was obtained and stored as previously described [44]. No clinical information was obtained from this cohort of patients. FFPE GBM tissue or frozen non-neoplastic tissue was unarchived and processed by our institutional Tissue Acquisition and Cellular/Molecular Analysis Shared Resource (TACMASR). IDH1 R132H mutational status was determined during clinical care using either a commercial immunohistochemistry (IHC) stain (Ventana Medical Systems, Tucson, AZ, USA) or the CLIA certified reference lab at the Mayo Clinic (Rochester, MN, USA). MGMT promotor methylation status was determined as part of their clinical care for only 2 patients using the CARIS platform (CARIS Life Sciences, Phoenix, AZ, USA). For the remaining patients, tissue samples were sent to the Mayo Clinic for methylation-specific polymerase chain reaction (PCR) analysis to determine their MGMT promotor methylation status. This retrospective study was deemed to meet the criteria for exemption under 45 CFR 46.101(b) by the Institutional Review Board (IRB) and Human Subjects Protection Program of the University of Arizona. This decision was filed under protocol #1709802216.

### 4.2. NanoString Gene Expression Data Collection and Analysis

Archived FFPE GBM tissue was de-paraffinized and RNA was isolated using a Roche HighPure FFPET RNA Isolation spin-column kit. Frozen non-neoplastic temporal lobe tissue (archived in RNAlater storage reagent) was lysed and homogenized, and RNA was isolated by organic extraction, followed by purification using a Qiagen RNeasy spin-column kit. Subsequently, 300 ng of purified RNA from FFPE tissue or 100 ng of purified RNA from frozen non-neoplastic brain tissue was hybridized with the gene expression code set probes of an nCounter PanCancer Human Pathways panel (NanoString Technologies, Seattle, WA, USA). Isolation and binding of hybridized probes to an optical cartridge was performed on an automated nCounter Prep Station (NanoString Technologies). The cartridge was then scanned by means of an nCounter Digital Analyzer (NanoString Technologies) to obtain gene-specific probe counts. All sample processing and data collection was performed by the University of Arizona Genetics Core facility.

Raw data were analyzed by means of NanoString Technologies’ nSolver Analysis (version 4.0) software, equipped with the supplementary Advanced Analytics package (version 2.0). Data were normalized against internal positive and negative controls, as well as 40 housekeeping genes, using the geNorm algorithm [45] per the manufacturer’s instructions [46].

### 4.3. Statistical Methods

The nSolver software (version 4) calculated differential expression (DE) of individual genes, with adjusted *p*-values to account for multiple comparisons with the Benjamini–Yekutieli method [47]. In addition to DE comparisons, the nSolver software also generates pathway-based summary statistics known as pathway scores and directed global significance scores. Pathway scores are calculated as the first principal component of the pathway genes’ normalized expression based on an algorithm developed by Tomfohr et al. [48]. Directed global significance scores measure the tendency of a pathway to have over or under-expressed genes by calculating the square root of the mean squared t-statistic of each gene in a pathway with an additional term to take the sign of the t-statistic into account.

Patient characteristics were compared between shorter and longer than median OS groups or between our entire institutional dataset and the TCGA dataset by means of Fisher’s exact test and *t*-test as indicated, using the open-source software R (version 3.3.1) and R Studio (version 1.0.136) [49]. Cox Proportional Hazards (CPH) models and Kaplan-Meier (KM) survival estimates were also generated in R using the survival and survminer packages. There were two patient samples whose MGMT status was reported as ‘Indeterminate’ by the reference laboratory and for the sake of the CPH models, their MGMT methylation status were categorized as negative, as their measured MGMT expression levels were more consistent with the unmethylated cohort.

Finally, a weighted average of the clinical and genomic factors was calculated to generate a single prognostic factor, which we call the glioblastoma prognostic index (GPI). Any clinical factor or gene expression level that was associated with OS with a *p* < 0.1 on univariate CPH analysis was included in a multivariable CPH model to generate regression coefficients that were then used as the weighting factors for the GPI score.

### 4.4. TCGA Validation

In order to validate our data with a larger dataset, we obtained clinical data, including IDH1 mutational status and MGMT promotor methylation status, for 543 patients analyzed as part of The Cancer Genome Atlas (TCGA) project from the cBio Portal [50,51] (www.cbioportal.org accessed on 1 June 2022). Unfortunately, this source of clinical data did not include reliable information on the extent of resection for these TCGA patients. Gene expression data from these TCGA patients for the four genes used to calculate the GPI were obtained from the University of California, Santa Clara (UCSC) Xena data portal [52], from an experiment conducted by the University of North Carolina TCGA genomic characterization center with an Agilent 244K custom gene expression microarray (G4502A_07_2). To make the two datasets as similar as possible, we excluded patients less than 18 years of age, those with IDH1 mutations, or those with an unknown MGMT promotor methylation status. This left 188 patients to be analyzed in this external validation cohort. To account for the differences in the dynamic range of mRNA counts between the two quantification methods used in our NanoString quantified cohort versus the microarray-quantified TCGA cohort, the gene expression levels of each cohort were normalized using a Z-score transformation and a new Z-score based multivariate CPH model was calculated for our cohort to derive the modified regression coefficients needed to generate a comparable GPI for the 188 TCGA patients. The multivariate CPH model used to generate the regression coefficients that was used to calculate the GPI was then repeated with these Z-score transformed gene expression values. For the sake of the model, an assumption was made that each of the TCGA patients had a GTR of their tumor at the time of their surgery. A new Z-score based formula was then used to calculate a GPI for each of the TCGA patients. Because the sample size was much larger with the TCGA dataset, rather than use the median GPI to divide the cohort into 2 equally distributed sub-groups, we calculated the optimal GPI cut point using maximally selected rank statistics [53], allowing for no more than 60% of the cohort to be in a single GPI group.

## 5. Conclusions

In summary, the work reported here has defined a panel of clinical and genomic factors that significantly refines OS prognosis in IDH1 wildtype GBM patients. While these results are intriguing, they need to be confirmed with larger sample sizes, using the nCounter platform and preferably in a prospective manner, to validate the GPI as a clinically useful prognostic tool in GBM. These results also suggest that further investigations of COL1A2, NGFR, IFGBP3, and WIF1 are warranted to better understand their roles in GBM pathogenesis and/or treatment resistance.

## Figures and Tables

**Figure 1 ijms-25-03668-f001:**
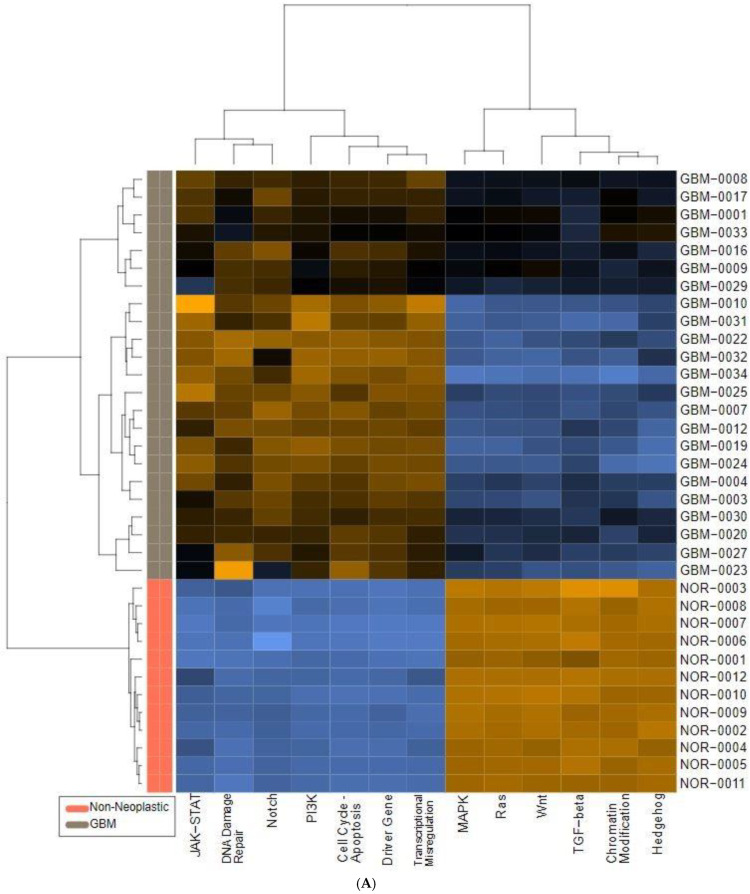
Pathway score comparisons. Heatmap of pathway scores between samples of GBM and non-neoplastic brain tissue (**A**) GBM tissues with Long OS versus those with Short OS (**C**). Samples are in rows while pathways are shown in columns and the warmer the color (red) the more upregulated the pathway is, while the cooler the color (blue) the more downregulated the pathway. (**B**,**D**) show the average pathway scores for each group for each comparison.

**Figure 2 ijms-25-03668-f002:**
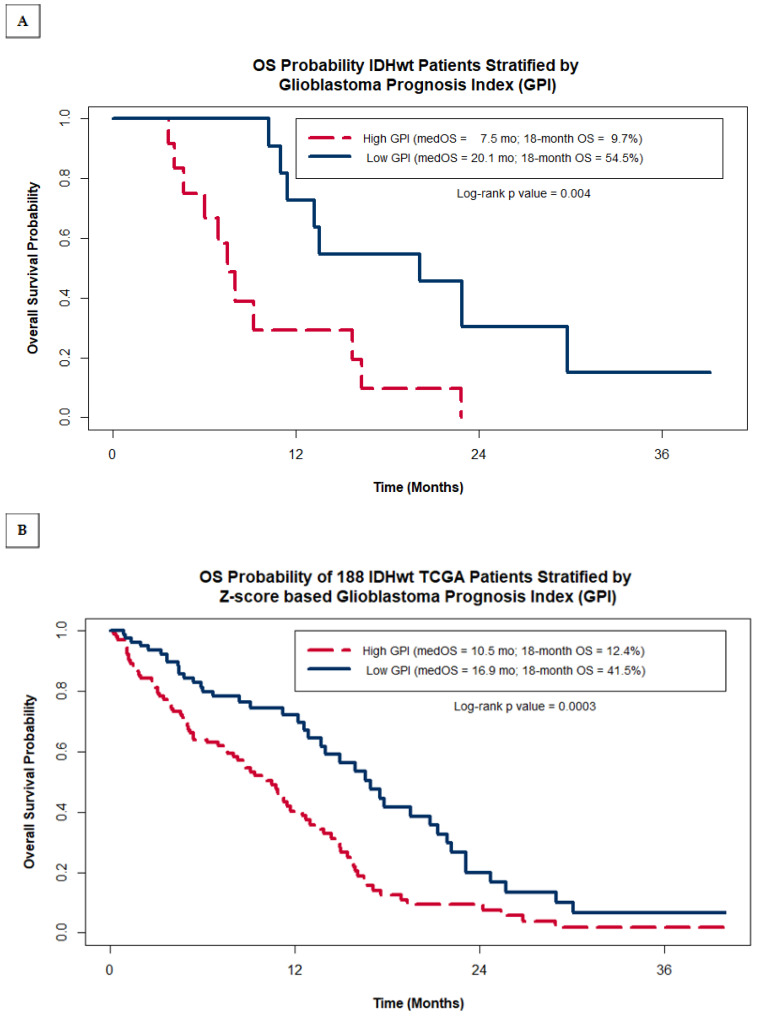
Kaplan–Meier OS probability estimates comparing patients grouped by GPI within our original institutional cohort of 23 IDHwt GBM patients (**A**) and within the entire cohort of 188 IDHwt TCGA patients (**B**). (**C**) shows the Kaplan–Meier OS probability estimates for the TCGA cohort stratified by MGMT promotor methylation status and GPI.

**Figure 3 ijms-25-03668-f003:**
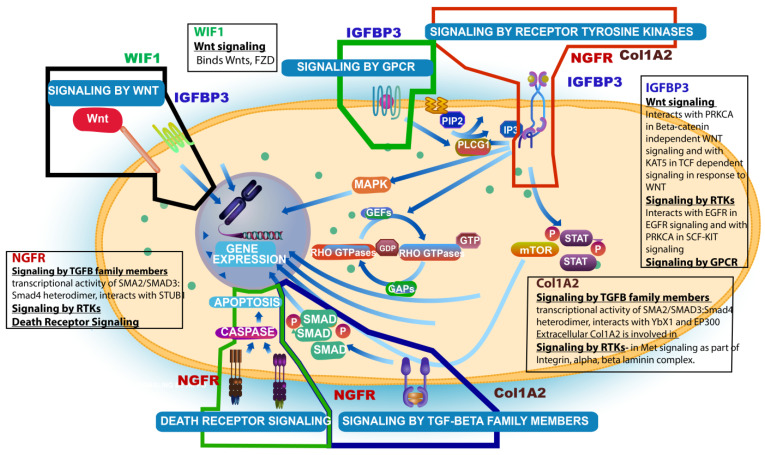
Summary schema of interactions of the 4 genes (NGFR, IGFBP3, WIF1, and Col1A2) involved in the derivation of the Glioblastoma Prognostic Index (GPI) and respective signaling pathways affected (main pathway figure was taken from Reactome Resources).

**Table 1 ijms-25-03668-t001:** (A). Patient, tumor, and treatment characteristics; (B). Comparison of our institutional cohort and TCGA cohort.

**A**			
	**Short OS** **≤ Median**	**Long OS** **> Median**	**Comparison** * **p** * **-Value**
Number of Patients (pts)	12	11	
Males/Females (numberof pts)	9/3	8/3	1
Median Age at Dx (Range)	70.5 (41–80)	63 (43–77)	0.19
Median KPS (Range)	85 (60–90)	80 (50–90)	0.47
Avg Tumor Size in cm^2^ (Range)	20.8 (5.4–36.2)	20.5 (1.9–41.2)	0.94
MGMT Promotor Methylation (number of pts)	2	3	0.64
Extent of Surgery (number of GTR/ STR)	2/10	5/6	0.19
Completion of concurrent TMZ (% of pts)	91.7	100	1
Range of RT Dose (Gy)	60–74	60–75	
**B**			
	**NanoString** **Cohort**		**TCGA Cohort**
Number of Patients	23		188
Males/Females	17/6		109/79
Median Age at Dx (Range)	64 (41–80)		63.5 (23–88)
Median KPS (Range)	80 (50–90)		80 (20–100) *
Avg Tumor Size in cm^2^ (Range)	20.7 (1.9–41.2)		Unknown
Median OS (Months)	11.4		8
MGMT Promotor Methylation Status (number/%)			
Unmethylated	18/78.3%		104/55.3%
Methylated	5/21.7%		84/44.7% ^&^
Extent of Surgery (number/%)			
GTR	7/30.4%		unknown
STR	16/69.6%		
Completion of concurrent TMZ (% of pts)	95.7		
Range of RT Dose (Gy)	60–75		

* KPS data missing in 25.5% of TCGA patients; ^&^ Significantly more MGMT methylated patients in the TCGA cohort (chi-squared *p* = 0.044).

**Table 2 ijms-25-03668-t002:** Differential expression of individual genes.

Gene	Log2 Fold Change: Non-Neoplastic vs. GBM	*p* Value	Log2 Fold Change: GBM Long vs. Short Survival	*p* Value	Pathway
*HIST1H3G*	8.3	7.64 × 10^−18^	1.29	0.00537	Transcriptional Misregulation
*MMP9*	8.29	7.61 × 10^−14^	−1.74	0.0268	Transcriptional Misregulation
*IBSP*	7.91	5.14 × 10^−13^	1.74	0.0212	PI3K
*COL1A2*	7.07	7.97 × 10^−13^	2.55	0.00266	PI3K
*IGFBP3*	3.88	1.67 × 10^−11^	1.67	0.00119	Transcriptional Misregulation
*NGFR*	3.84	3.39 × 10^−8^	1.9	0.00656	PI3K, Ras, Transcriptional Misregulation
*COL4A6*	2.85	1.95 × 10^−7^	−1.36	0.0473	PI3K
*PTTG2*	1.93	3.10 × 10^−6^	1.02	0.0319	Cell Cycle—Apoptosis
*MMP7*	1.82	0.000651	1.68	0.00867	Wnt
*BMP2*	1.29	4.94 × 10^−5^	−1.04	0.00337	Hedgehog, TGF-beta
*WNT7B*	−1.44	0.00306	−2.77	0.00165	Hedgehog, Wnt
*WIF1*	−4.3	2.22 × 10^−8^	−1.7	0.0398	Wnt

**Table 3 ijms-25-03668-t003:** Global significance scores as defined by the nCounter PanCancer Human Pathways panel.

Directed Global Significance Scores		
Pathways	Non-Neoplastic vs. GBM	GBM Long vs. Short Survival
Notch	6.507	−1.159
Transcriptional Misregulation	6.265	0.494
Cell Cycle—Apoptosis	5.407	−0.713
DNA Damage—Repair	5.324	0.491
Driver Gene	4.73	−0.862
PI3K	4.585	−0.753
JAK-STAT	4.34	−0.939
Hedgehog	2.36	−0.919
TGF-beta	1.488	−1.184
Ras	−2.103	−0.899
MAPK	−2.382	−0.859
Chromatin Modification	−2.889	−1.09
Wnt	−2.99	−0.893

**Table 4 ijms-25-03668-t004:** Cox proportional hazards models. Cox proportional hazards analysis of the clinical factors commonly associated with overall survival in GBM patients as well as the 6 genes that had more than 8-fold DE in the GBM vs. non-neoplastic comparison and more than 3-fold DE in the ‘Short OS’ vs. ‘Long OS’ comparison. The regression coefficients shown in the right column were then used to generate the GPI.

Univariate CPH Model			Multivariate Regression Coefficients
**Clinical Factors**	**HR**	***p* Value**	
Age	1.06	0.061	0.074
KPS Group (<70 vs. ≥70)	0.81	0.792	
Tumor Size (cm^2^)	1.02	0.423	
Extent of Resection (STR vs. GTR)	3.65	0.047	1.430
MGMT Promotor Status (Methylated vs. Unmethylated)	0.53	0.313	−0.876
**Gene Expression**			
*MMP9*	0.98	0.858	
*IBSP*	1.23	0.173	
*COL1A2*	1.38	0.034	0.243
*IGFBP3*	1.65	0.010	0.090
*NGFR*	1.37	0.043	0.448
*WIF1*	0.80	0.095	−0.101

## Data Availability

Data is contained within the article.

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
