# Peer review of "Gene Expression Patterns Associated with Survival in Glioblastoma"

_ijms, 2024, doi:10.3390/ijms25073668_

Round 1

Reviewer 1 Report

Comments and Suggestions for Authors

In this manuscript, for glioblastoma (GBM), the authors searched for biologic differences that might correlate with overall survival after standard treatment with surgery, chemo-radiation (chemo-RT), and adjuvant TMZ chemotherapy. Using comparative differential expression analyses combined with multivariate Cox proportional hazards (CPH) models, they have derived a prognostic index that can discriminate between GBM patients with significantly different OS expectations. They then validated the utility of this prognostic index within an independent, publicly available dataset from The Cancer Genome Atlas (TCGA). They concluded that this novel mRNA-based prognostic index could be useful in classifying GBM patients into risk groups and refine prognosis estimates to better inform treatment decisions or stratification into clinical trials. In summary, this manuscript was well-written, comprehensive, and the experiments were carefully designed and performed. The significance and novelty were clearly stated that the reported here has defined a panel of clinical and genomic factors that significantly refines OS prognosis in IDH1 wildtype GBM patients. After all, only some minor changes need to be addressed before final publication.

1. The Table formatting is not clear, could the authors change the table format to an easier-to-read one?

2. The Figure legends formatting: "A, B, C, etc." are not consistent, please fix it.

3. The patients sample pool is really low and not diverse, only 23 patients were selected. I know it is hard to get real patients data, but could the authors increase the amount of sample pool if possible. Also, please remember to thank these "cancer fighters" in the acknowledgement section. Their willingness to participant in this research study should be much appreciated. Thank you.

Author Response

Thank you very much for taking time to serve as a reviewer for the IJMS and for your thoughtful review of our manuscript.  We have addressed your concerns below.

Item 1.  Table formatting on all tables and figures have been reviewed and minor adjustments made as appropriate for clarification.

Item 2.  All figure legends have been standardized in formatting for consistency.

Item 3.  Thank you for your mention of acknowledgement of the cancer fighters who participated in the research for this study.  We have added an Acknowledgement paragraph thanking them appropriately.

We acknowledge the fact that we only had 23 patients in our institutional cohort. The reviewer correctly would like to see more patients. Unfortunately, this was retrospective review and we only had 23 patients who had been uniformly treated and for whom we had data. Because of these limitations, we accessed the TCGA data base (188 additional patients) which provided a validation of the findings from our small cohort of 23 patients.

Reviewer 2 Report

Comments and Suggestions for Authors

The group of authors used comparative differential expression analyses combined with multivariate Cox proportional hazards models, to devise a prognostic index that can discriminate between glioblastoma (GBM) patients with significantly different overall survival expectations. Then, they tried to validate the utility of this prognostic index within an independent, publicly available dataset from The Cancer Genome Atlas.

The topic is considered original as it investigates significant markers to classify GBM patients into risk groups and refine prognosis estimates to better inform treatment decisions or stratification into clinical trials. A higher number of samples are required for further studies. The conclusion is consistent with the presented evidence and the numbers of references are appropriate. The most important improvement is as follows:

The authors must draw several professional schematics or figures to show the function of the genes with the most significant differential expression.

Author Response

Item 1. We acknowledge the fact that we only had 23 patients in our institutional cohort. The reviewer correctly would like to see more patients. Unfortunately, this was a retrospective review and we only had 23 patients who had been uniformly treated and for whom we had data. Because of these limitations, we accessed the TCGA data base (188 additional patients) which provided a validation of the findings from our small cohort of 23 patients.

Item #2- professional schematics developed and included in the resubmitted manuscript.

Reviewer 3 Report

Comments and Suggestions for Authors

REVIEW

Glioblastoma multiforme (GBM) belongs to the primary tumours of the central nervous system (CNS), with a high degree of malignancy both histopathologically and clinically. It is assigned the highest malignancy grade IV according to the World Health Organization (WHO) classification. It accounts for 10–18% of all intracranial tumors and 50–60% of all stellar glial tumors. The annual incidence of GBM is about 5 cases per 100,000 people. The incidence in adults increases with age, with the highest incidence occurring in the 5th and 6th decade of life.

There are two forms of GBM — primary and secondary. The first of these occurs most often in people over 55 years of age. It is characterized by a rapid clinical manifestation and corresponds to tumors emerging de novo, which already in the initial period of growth present the weaving typical of GBM. In the secondary form, usually affecting people under 45 years of age, the tumor develops on the basis of a glioma with a lower degree of malignancy. In these cases, the transformation time to malignant and clinical manifestation may be 1-10 years (average 4-5 years). Both GBM subtypes also differ in molecular profile, which determines tumor transformation and prognosis. In primary glioma, a common abnormality is amplification or overexpression of epidermal growth factor receptor (EGFR) and MDM2 protein, loss of heterozygousity on chromosomes 10p and 10q, deletion of suppressor p16, which controls cell growth by inhibition of CDK4 and CDK6 kinases, and deletion of suppressor PTEN on chromosome 10. Secondary glioma is often associated with a mutation of the p53 protein gene, loss of heterozygousity on chromosomes 10q, 17p, 19q and overexpression of the platelet factor receptor.

The GBM macroscopic image is heterogeneous. Within the same tumour, variable histological weaving and large cell diversity with pronounced atypia and very numerous cell divisions are observed. The representative feature of the hyperplasia is its intrusive nature, characterized by angiogenesis with the formation of globular structures, uncontrolled proliferative activity, the presence of necrotic foci surrounded by pseudopalisadic zones and hemorrhagic foci with different time of onset. Glioma multiforme is sometimes so vascular that it resembles a vascular malformation or cerebral infarction with a zone of excessive blood flow. The tumor can develop in any area of the brain, but it is usually localized supernutally. It usually shows subcortical hyperplasia. It rarely causes metastases outside the CNS.

The aim of this study was to investigate changes in the expression of genes COL1A2, IGFBP3, NGFR and WIF1, which are associated with overall survival (OS) in glioma.

The title of the work reflects the scope and content of the work

Abstract

The abstract contains the most important results obtained in the study, I have no comments on this part.

Introduction:

The introduction to the subject of research and the selection of literature is appropriate.

It is worth putting a research hypothesis at the end, and the goal of the work requires re-editing.

Material and Methods:

The research methods used are modern and suitable for this type of research. The number of patients n=23 is unsatisfactory should be higher. The protocol of diagnostic tests lacks MRI tests and their comparison with the expression of the studied genes.

Results:

The results are presented in a comprehensible way and are presented in the form of 4 tables and 2 graphs with their description. This chapter is well written and unobjectionable.

Discussion:

The discussion is conducted in a fair manner. The authors referred to all the results obtained in the study. Relevant literature has been cited.

Final remark:

The submitted work for evaluation is of a very high scientific standard and very well edited. It should be noted that the authors have extensive knowledge in the field of genomic studies and, above all, in the assessment of the etiology and course of development of glioma tumour. However, the work has shortcomings in the field of MRI diagnosis and comparison with expression of COL1A2, IGFBP3, NGFR and WIF1 genes. Once this information has been completed, the work will be ready for printing in IJMS.

Author Response

Thank you very much for taking time to serve as a reviewer for the IJMS and for your thoughtful review of our manuscript.  We have addressed your concerns below.

Check list “Are the methods adequately described?”  We have reviewed the methods portion of the manuscript and all corresponding tables and figures and adjusted them to provide greater clarity and cohesiveness to the methodology of the study.

Item 1.  “It is worth putting a research hypothesis at the end of the introduction, and the goal of the work requires re-editing.” – this was an excellent suggestion, and a sentence was added to the Introduction.

Item 2.  “The number of patients n=23 is unsatisfactory should be higher.” We acknowledge the fact that we only had 23 patients in our institutional cohort. The reviewer correctly would like to see more patients. Unfortunately, this was retrospective review and we only had 23 patients who had been uniformly treated and for whom we had data. Because of these limitations, we accessed the TCGA data base (188 additional patients) which provided a validation of the findings from our small cohort of 23 patients.

 “The protocol of diagnostic tests lacks MRI tests and their comparison with the expression of the studied genes.” Patients had a baseline MRI following resection of the GBM and then followed with serial MRIs every 2 months. However, we only addressed the overall survival of the 23 patients and not their Disease-free Survival. A sentence regarding the imaging protocol was placed in the Methods.

Item 3.

Round 2

Reviewer 2 Report

Comments and Suggestions for Authors

The quality of Figure 3 and the font size inside the figure must be increased.

Author Response

please see new improved Fig 3 in the manuscript

thanks
